# The Association between Asthma and OSA in Children

**DOI:** 10.3390/children9101430

**Published:** 2022-09-21

**Authors:** Nicholas Garza, Manisha Witmans, Martina Salud, Pamela Gail D. Lagera, Vince Aaron Co, Mary Anne Tablizo

**Affiliations:** 1Valley Children’s Hospital, Madera, CA 93636, USA; 2Department of Pediatrics, University of Alberta, Edmonton, AB T6G 2B7, Canada; 3Ateneo University School of Medicine and Public Health, Pasig 1604, Philippines; 4University of California San Francisco Parnassus Campus, San Francisco, CA 94143, USA; 5Department of Biology, California State University Fresno, Fresno, CA 93740, USA; 6Department of Pediatrics, Stanford University School of Medicine, Palo Alto, CA 94304, USA

**Keywords:** asthma, OSA, pediatrics

## Abstract

Obstructive sleep apnea (OSA) and asthma are two of the most prevalent and commonly co-existing respiratory conditions seen in the pediatric population. Studies linking asthma and OSA in children are limited but indicate that there is a bi-directional relationship between them with significant overlap in the symptoms, risk factors, pathophysiology, comorbidities, and management. It is suggested that there is a reciprocal association between asthma predisposing to OSA, and OSA worsening symptom control and outcomes from asthma. It stands to reason that inflammation in the upper and/or lower airways can influence each other. Most of the pediatric literature that is available evaluates each aspect of this relationship independently such as risk factors, mechanisms, and treatment indications. This article highlights the relationship between OSA and asthma in the context of shared risk factors, pathophysiology, and available management recommendations in the pediatric population. Early recognition of the co-existence and association between OSA and asthma could ideally improve the treatment outcomes for these two conditions. Gaining a better understanding of the mechanism of this relationship can help identify nuances for medical management, optimize treatment and protect this population at risk from associated morbidity.

## 1. Introduction

Asthma and obstructive sleep apnea (OSA) are two chronic airway-related disorders that share common risk factors. Asthma is a globally significant chronic disease for both children and adults, with asthma incidence and prevalence being highest in children [1], affecting almost 7,000,000 children in the US [2]. The occurrence of asthma varies from 3–5% in developing countries to greater than 20% in developed countries. Childhood asthma remains a huge global burden with many hospital admissions for exacerbations especially during fall every year [3,4]. On the other hand, the prevalence of pediatric OSA is rising with a reported range of 1–6% overall [2]. Pediatric OSA occurs most commonly in children ages 2 to 8 years old, associated mostly with tonsillar and adenoid hypertrophy. Both patients with asthma and OSA have reported poor sleep quality. Children with asthma and OSA have been shown to have sleep fragmentation and difficulty with sleep maintenance (frequent wake periods in the night), as well as daytime sleepiness and fatigue [5]. 

Although there are few studies done on the relationship between pediatric OSA and asthma, the literature so far points to a likely bidirectional relationship with significant overlap in the symptoms, the interaction between risk factors, comorbidities, and management [6,7,8]. Castro-Rodriguez and colleagues performed an elegant review of the literature to determine causality between asthma and OSA in children using Bradford Hill criteria [9]. They highlighted the association between asthma and OSAS but could not prove causality because of the lack of temporality and directionality based on the reported literature. Further studies continue to report associations between the two highly prevalent conditions. However, the studies are often limited to evaluating questionnaires and limited objective data for diagnosis of the conditions. With their common symptoms and shared risk factors, it is not known whether these are two truly separate conditions that share similar symptoms or if these two conditions share common pathophysiological mechanisms. This review discusses these two systemic inflammatory, chronic conditions by addressing common risk factors, pathophysiology, available management recommendations, and evidence that supports the possible relationship between OSA and asthma in the pediatric population.

## 2. Pathophysiology on Bidirectional Link of Asthma and OSA

Asthma is a common chronic inflammatory condition of the lung and airways that causes episodic, often reversible airflow obstruction. Inflammation results in airway hyperresponsiveness (AHR) to common exposures such as respiratory infections, allergens, exercise, irritants, weather changes, emotions, and other triggers. Patients will commonly present with symptoms of cough, shortness of breath, chest tightness, and/or wheezing due to airflow limitation [3]. Several pathologic processes are involved in the chronic inflammation seen in asthmatic patients. At the cellular level, inflammatory cells are recruited to the lung airways, eliciting epithelial damage and desquamation in the lumen of these airways. At the molecular level, these inflammatory cells secrete cytokines (IL-4, IL-5, IL-9 and IL-13), which are the intermediate agents that orchestrate the inflammatory process [10]. Cytokines, such as IL-5 is necessary for eosinophilic inflammation and IL-4 play a critical role in the isotype-switching of B cells to produce IgE instead of IgG, with IgE being a key antibody in allergen sensitization to certain environmental triggers [10]. Another cytokine is IL-13, which has been found to induce AHR and mucus hypersecretion by activating the STAT6 pathway in the airway epithelium. Numerous targeted biologic therapies now aid asthma management through cytokine and IgE regulation. Bronchoconstriction of airways as a result of the cascade of inflammation-causing smooth muscle contraction is another pathological process involved in asthma. 

OSA results from a partial or complete upper airway obstruction and/or intermittent complete obstruction that can disrupt air exchange during sleep, resulting in perturbations in oxygenation and ventilation. OSA falls on the severe end of the sleep-disordered breathing (SDB) spectrum [11]. Because OSA disrupts nocturnal respiration and sleep architecture, there is an increase in the number of arousals per hour of sleep (arousal index), which can lead to sleep fragmentation. The disruption in nocturnal respiration can also lead to hypoxemia and hypoventilation, which may serve as potential stimuli for the release of vasoactive substances and impair endothelial dysfunction. Endothelin, a marker of endothelial dysfunction, is increased in sleep-disordered breathing [12]. Cytokines that have been linked to asthma are also noted in children with OSA.

Studies have shown that SDB has a statistically significant association with asthma [13]. Current evidence suggests shared airway and systemic inflammatory pathways for both asthma and OSA [6]. Upper airway inflammatory processes may play a role in further exacerbating lower airway inflammation. On the other hand, lower airway inflammation may also increase the risk of adenotonsillar proliferation increasing the risk of OSA [14]. This dynamic relationship has been referred to as the “one-airway” hypothesis. In this concept, hypoxia and inflammation from upper airway obstruction are associated with disease severity and phenotype of the lower airways. The reciprocal inflammatory response in both upper and lower airways correlates with atopy and increased leukotriene response. The result is constriction of the lower airways, increased nasal resistance, and increased pharyngeal collapsibility [15,16,17]. 

Multiple studies have shown that asthma is an independent risk factor for the development of OSA. There are proposed mechanisms for predisposition to having OSA in asthmatic patients. Asthma is associated with chronic inflammation, which could affect the force generation properties of the respiratory muscles, including the upper airway dilator muscles predisposing patients to have OSA [6,18]. Bronchoconstriction in asthma causes airway collapse and increased vagal tone. This reduces total airway cross-sectional area and is associated with expiratory oropharyngeal and glottic constriction increasing the risk of developing OSA [2,5,6]. The airway collapse can trigger the release of inflammatory markers [2]. The inflammatory response can lead to increased neutrophil and T-cell lymphocytic infiltrate recruitment in the airways and an increase in T-cell lymphocyte infiltration in the upper airway mucosa increasing predisposition to OSA. It has also been proposed that asthma can affect the integrity and patency of the upper airways through sleep deprivation and fragmentation causing systemic inflammation and subsequently weakening of respiratory muscles. This weakening reduces the overall patency of the upper airways by increasing upper airway collapsibility [2,6,19]. While asleep, asthmatics also tend to have a low functional residual capacity which would compromise the stabilizing effect of the tracheal pull on the pharyngeal portion of the upper away segment [6,20]. 

Furthermore, the link between OSA and asthma is interchangeable with several proposed pathways that are observed to contribute to asthma pathogenesis in OSA. OSA can increase the burden on already challenged asthmatic airways by increasing resistive load on the upper and lower airways. One proposed mechanism involves elevations in the physiologic vagal tone. OSA can induce upper airway vibration and collapse of the airways. This vibration and collapse can lead to bronchoconstriction by increasing the airway vagal tone. Increased vagal tone exacerbates nocturnal asthma. Nocturnal hypoxia and distally transmitted mechanical stress from snoring may also exacerbate the lower airway inflammation in asthmatic patients. Additionally, intermittent hypoxia from OSA could impair arousal thresholds to resistive loading, cough, and asthma symptom perception, which are all important defense mechanisms for asthmatic patients during sleep [2,6]. OSA can also increase the chances of gastroesophageal reflux, which is a well-known trigger of asthma [2]. 

## 3. Common Risk Factors 

### 3.1. Obesity

Obesity itself is a pro-inflammatory state and has been shown to be associated with comorbidities related to different organ systems in humans. In both OSA and asthma, obesity plays a key risk factor. Body Mass Index (BMI) is a significant risk factor in the occurrence of asthma with a positive linear relationship [21]. The obese asthma syndrome is multifactorial with mechanisms that are suggested to be related to genetics, nutritional factors, inflammatory pathways, metabolic disorders, and transformations in lung physiology and anatomy [21]. Increased fat accumulation in the thoracic and abdominal cavities leads to airway and lung restriction and a corresponding reduction in lung volume [22]. Diets rich in saturated fatty acids increase neutrophil recruitment within the airway [23]. Macrophage activation by innate lymphoid cells has been expressed as higher sputum counts amongst individuals with atopy-obesity overlap, which suggests a strong association of the innate pathways for obesity-related asthma [24]. Obesity is a state of chronic systemic inflammation that shares underlying mechanisms with asthma, fueling the ongoing debate regarding the direct causality of the two entities [25].

Elevations in the BMI is a way to quantitatively categorize obesity and approximately 46–59% of children with elevated BMI have OSA. Adenotonsillar hypertrophy and fat deposition in the lateral fat pads, uvula, and tongue are contributing anatomic factors to airway narrowing that can cause OSA. Obesity is an independent risk factor for OSA in children after controlling for adenotonsillar hypertrophy [12,26]. It is possible that the same mechanism involving a diet high in saturated fatty acids that increase neutrophil recruitment within the airway inflammation and increase macrophage activation can explain the same effect not only in the lower but also in the upper airway causing OSA in obese patients [23]. Furthermore, any additional pressure from increased body mass on the airway results in decreased compliance and increased upper airway collapsibility. Over the past three decades, the rising prevalence of childhood obesity has contributed to increasing the risk and severity of OSA in children. This would lead us to presume improvements in the prevalence of childhood obesity would decrease the burden of OSA.

### 3.2. Prematurity 

Prematurity and very low birth weight can be additional neonatal risk factors for the development of both asthma and OSA. Prior studies have suggested associations between prematurity and OSA. One study has shown that a history of prematurity is associated with increased AHI and diagnosis of severe OSA compared to children born at term [27]. In fact, the risk of OSA in children who are born prematurely is increased by 4–5 fold. The narrowed airway dimension and upper airway diameter are two of the proposed reasons for the increased risk of OSA in children born prematurely. The craniofacial profile of premature infants with a high and narrow arch with maxillary deficiency increases the risk of OSA. 

Premature newborns often require supplemental oxygen and positive pressure support. These life-saving neonatal interventions, however, may also increase the risk for future wheezing and abnormal lung mechanics. The airway growth and dimensions in premature infants are generally decreased compared to infants born at term which may increase the risk of development of asthma based on lung mechanics. It is proposed that intermittent hypoxemia and oxygen exposure during the early postnatal window of vulnerability predisposes the infant to longer-term airway hyperreactivity. The decreased airway growth, abnormal lung function, and intermittent hypoxemia are among the common abnormalities that increase the risk for both asthma and OSA [20,28]. 

### 3.3. Environmental Factors

Environmental tobacco smoke (ETS) exposure and air pollution are known risk factors in the development of OSA. This may be explained by irritation or inflammation of the upper airway mucosa and increased airway resistance leading to pharyngeal collapsibility [29,30]. The association between asthma and ETS exposure has been well studied and is likely related to ETS-induced inflammatory response in the lower airway mucosa [31]. 

### 3.4. Lower Respiratory Tract Infection in Early Life

Children with a history of severe lower tract infection (LRTI) in early life have a significant risk of developing OSA and asthma. A study analyzing the clinical data on 114 children for the first 5 years of life showed that early LRTI increased the risk of pediatric OSA independent of other covariates and risk factors. This study also showed that children who had severe respiratory syncytial virus bronchiolitis during infancy had 2 times higher odds of OSA at 5 years in comparison with children without exposure [32]. Lower respiratory tract infection during infancy is also a known risk factor for developing asthma in children. This could be related to altered lung mechanics occurring from the LRTI early in life or that the initial predisposition made the child vulnerable to the LRTI [33]. 

### 3.5. Allergic Rhinitis

Allergen sensitization from environmental exposures at a young age is another notable association with the development of allergic disease and childhood asthma [24]. Factors such as host genetics, host immunological factors, timing, and other exposures likely affect how the environment might predispose someone to develop allergies. Allergic rhinitis (AR) is under the spectrum of allergen sensitization, also known as atopic disease. AR is a major risk factor for developing asthma [34]. Studies have shown that up to 20–50% of those with AR can have asthma and as much as 60–78% of asthmatics have AR [35]. This risk of developing asthma increases with the increasing severity of rhinitis [36]. In one study among children less than 7 years old, there was a 3-fold increased risk of developing asthma compared to subjects without AR. It is suggested that the nose and bronchi are linked through systemic exposure hence allergic inflammation that started in the upper airway can also involve the lower airways [37].

AR has also been associated with sleep-disordered breathing (SDB). Because the nose represents the primary route of breathing during sleep, AR and chronic rhinosinusitis can influence altered nocturnal oropharyngeal mechanics which can cause or contribute to OSA. Increased nasal resistance and associated nasal inflammation from AR can result in more negative oropharyngeal pressure during inspiration, predisposing to upper airway collapse [5,6]. One study has shown that there is an increased frequency of adenoidal hypertrophy in children with AR [38]. Patients with persistent AR and asthma tend to predominantly breathe through their mouth which can predispose children to craniofacial changes including narrowed oropharyngeal diameter leading to OSA. Although AR has been linked to a higher prevalence of SDB, most of the studies were done using questionnaires and without polysomnographic evidence to confirm the diagnosis of OSA. Thus, further studies are needed to better assess the relationship between these two conditions [39]. 

### 3.6. GERD

OSA promotes GERD, which is also a well-recognized trigger of asthma [2,6,8]. In a meta-analysis that proved a causative relationship between OSA and GERD, it was reported that episodes of airway acidification were very common in patients with sleep apnea [40]. CPAP utilization has been shown to decrease GERD in OSA patients. 

GERD has also been considered a possible risk factor for the development of asthma possibly due to chronic irritation of the airway tract following exposure to gastric acidity [41]. Repeated coughs in asthma can also increase the risk for GERD [41]. 

### 3.7. Sex 

OSA also does show gender predisposition differences much like asthma in certain stages in the human lifespan. Overall, asthma and OSA are more prevalent among boys than girls, especially in younger children. It has been proposed that the reason asthma occurs more often in males is due to the boys having a reduced airway size compared to girls until reaching maturity [42]. The tendency of OSA persists throughout life for males. Females tend to have an increased prevalence of asthma in puberty and beyond. OSA is more common in males than in females and this tendency tends to persist till after menopause when women have a much higher prevalence of OSA. 

## 4. Common Diagnostic Findings

OSA and asthma can have overlapping nocturnal symptoms and polysomnographic findings. Both OSA and asthma in children can be described by parents as having labored breathing during sleep, persistent snoring, nocturnal cough, and restless sleep. Asthma and OSA can also present with excessive daytime sleepiness and fatigue from fragmented sleep [43,44]. Some of the symptoms that are treated as asthma may be symptoms of OSA. On the other hand, cough and difficulty breathing in patients diagnosed with OSA may have underlying components of asthma. Knowing when the two conditions co-exist and performing appropriate diagnostic tests such as polysomnography, spirometry, or pulmonary function tests to identify each condition is important.

Evidence on the effects on sleep architecture and nocturnal oxygen saturation when asthma and OSA co-exist in children is emerging. One study in adults showed that the combination of asthma and OSA was associated with lower mean oxygen saturation, especially in REM sleep [45,46]. The combination of asthma and OSA was also associated with shorter sleep time, longer sleep latency, and longer sleep time in stage 1 and stage 2 but decreased percentage of time spent in sleep stage 3 and REM [43,44,47].

Children with asthma have poor sleep quality because of cough and wheeze [47]. Poorly controlled asthmatics have increased nocturnal awakenings and take a longer time to fall back to sleep [47]. Multiple logistic analyses revealed that wheeze was a risk factor for difficulty falling asleep, restless sleep, daytime sleepiness, and tiredness [29]. The sleep quality in children with OSA has the same pattern of increased nocturnal awakenings from obstructive events and subsequent arousals. As a consequence of poor sleep quality, children with OSA experience increased irritability and behavioral issues. 

One study on non-obese asthmatic children aged 6–12 years old with AHI < 5, showed longer sleep latency, increase leg movement index, and a lower ratio of slow wave sleep compared with non-asthmatic with AHI < 5. In the same study, the asthmatic group with AHI > 5 showed a higher ratio of REM, sleep stages 1 and 2, a lower ratio of stage 3, and increased respiratory arousal index and leg movement compared to non-asthmatic with AHI > 5 [48]. This study shows that sleep disturbance was found in asthmatic children with or without moderate to severe OSA [48]. In another study that used polysomnographic data to evaluate the sleep of asthmatic children with OSA, they saw a significant decrease in the quality of sleep and increased sleep fragmentation in asthmatic snorers than non-asthmatic snorers [49].

Intermittent nocturnal hypoxemia is a hallmark of OSA and can also be seen in severe asthma, or nocturnal asthma. Asthma-induced nocturnal hypoxemia can increase the severity of OSA and OSA-induced nocturnal hypoxemia can worsen the severity of asthma [2,6,8]. Fractional exhaled nitric oxide (FeNO), a surrogate biomarker used to detect airway inflammation, is increased in asthma. FeNO has also been shown to be increased in OSA [50].

There are no guidelines as to when to perform a sleep study in asthmatic children or when to perform an asthma work-up on OSA patients. In addition to the clinical presentation, diagnosis can be differentiated if it is from asthma alone if oxygen desaturation is not associated with evidence of increased upper airway resistance or overt obstructive events during polysomnography [50]. On the other hand, performing pulmonary function tests will show airflow obstruction in asthma but it is uncertain whether this airflow obstruction is also present in OSA.

## 5. Clinical Studies That Link Asthma and OSA 

According to studies, children with asthma are more at risk of snoring compared to children without asthma [2]. Polysomnography in children with asthma has shown a higher prevalence of snoring than in the general population [2]. One study showed that snoring and nocturnal awakening were more prevalent in children with wheezing (46% and 30%, respectively) compared to children without wheezing (26% and 20%). In 2014, a meta-analysis for the association between asthma and SRDB in 45,155 children (mean age 8.6 + 2.5) was done and showed that SRDB was significantly more frequent in children with asthma compared with non-asthmatics (23.9% vs. 16.7&, *p* < 0.001). These studies however were mostly based on questionnaires with only 2 studies that confirmed the diagnosis of OSA by polysomnography [51]. OSA is also associated with greater asthma severity, decreased asthma control, and increased risk for asthma exacerbations [6,8]. Children with poorly controlled asthma were also found to likely have OSA when compared to children without asthma [8]. In a study using polysomnographic data to define OSA in poorly controlled asthmatics undergoing routine clinical care, researchers were able to show that children with OSA had significantly worse asthma control scores, and treatment of OSA resulted in improved outcomes. This study also shows that there is a high frequency of OSA in children with moderate to severe asthma [14]. A retrospective study done on >500,000 children ages 2–18 years old hospitalized for acute asthma exacerbations concluded that OSA was an independent risk factor for increased utilization of invasive and non-invasive mechanical ventilation independent of obesity; as well as an increased total length of hospital stay and cost of hospitalization [52]. Kilaikode et al., also showed the need for starting CPAP therapy was higher in patients with both asthma and OSA than in those with OSA alone. Additionally, the study demonstrated that excessive daytime sleepiness in children with severe OSA was worse in asthmatics when compared to children without asthma [53]. 

## 6. Outcome of Treatment for OSA in Asthma Control

No clear guidelines exist for treatment when both OSA and asthma exist simultaneously, however, if the association is proven to be clear, it seems logical that control of one disorder will show benefits on the other and possibly vice versa. 

Tonsillectomy and adenoidectomy (TA) is the primary treatment for pediatric OSA. TA relieves the upper airway obstruction. By reducing the allergic and inflammatory burden of OSA on the upper airway should in turn benefit asthma control. There are retrospective database studies that have shown that TA is associated with improvements in asthma treatment outcomes [8,14,54,55]. It has also been shown that the persistence of OSA post-adenotonsillectomy can be related to the presence of comorbid asthma in children [54]. A large study, consisting of 13,500 children with OSA and asthma, found that TA resulted in a 30–40% reduction in asthma exacerbations and status asthmaticus episodes. This study also showed reductions in asthma-related presentations to the emergency department by 26% and hospitalizations by 36% [54]. In another study by Kheirandish-Gozal et al. of 92 poorly controlled asthmatics, 58 patients were found to have OSA confirmed by polysomnographic data. In this study, they found that at the 1-year follow-up, the asthmatics who had TA showed a significant decrease in acute asthma exacerbations requiring either less systemic steroid use or emergency visits. It also decreased the need for β-agonist inhaler use in patients with poorly controlled asthma [14]. In a retrospective study done by Busino et al., asthmatic children with OSA who had TA showed a reduction in the hospital visits, need for corticosteroids and improved asthma control scores [52,56].

CPAP is less commonly used in pediatric cases of OSA compared to adults but can be useful in cases such as refractory OSA, obesity, craniofacial malformation, or complex patients when surgery is not feasible. A study on the use of CPAP on adolescents with OSA and nocturnal asthma showed significant improvements in nocturnal asthma symptoms [8]. Based on these studies, there is evidence that treatment of OSA can improve asthma control. Although it makes sense that controlling asthma can improve OSA, to our knowledge, studies indicating this has not been published in the pediatric population. Further studies are needed to show that asthma treatment can improve the severity of OSA. 

Leukotrienes also play a key role in the pathogenesis and treatment of both asthma and OSA. Leukotrienes are known to be inflammatory mediators in asthma and leukotrienes have also been shown to be increased in the adenotonsillar tissue of children with OSA. Leukotriene receptor antagonists, such as montelukast, are common treatments for asthma. Both short-term monotherapy and long-term monotherapy have been shown to provide protection in asthmatic children [57]. Furthermore, long-term monotherapy of montelukast use on 689 preschool children with multi-trigger wheezing and persistent symptoms showed that daily montelukast given over a course of three months led to more symptom-free days as opposed to its placebo [58]. Because leukotrienes are found in adenotonsillar tissue, leukotriene receptor antagonists are also offered as one of the treatment options for mild OSA and can decrease the need for surgery in some OSA patients by decreasing airway inflammation [2]. One study demonstrated that a 4-month course of a leukotriene receptor antagonist correlated with decreased severity and symptoms of OSA when compared to a control group [2]. The treatment of inflammation of the airway using leukotrienes receptor antagonists be it asthma or OSA suggests a common pathophysiological process amenable to the same treatment occurring at different locations and influencing each other. 

Intranasal steroids are a common intervention in allergic rhinitis in children and work by decreasing nasopharyngeal inflammation and increasing airway diameter. Their ability to alter respiratory airway mechanics can possibly lead to changes in other respiratory pathologies. When used in asthmatics with allergic rhinitis, studies have shown a significant reduction in the severity of asthma symptoms and an increase in quality-of-life scores [41]. In a small cohort of 25 preschool-aged children, one study showed that regular use of intranasal fluticasone decreased OSA severity [41]. Cochrane review in 2020 assessed the efficacy of anti-inflammatory medications for objectively diagnosed OSA (AHI > 1) in 240 children (1 to 18 years). The study included 5 randomized controlled trials with follow-ups from 6 weeks to 4 months. The study showed possible short-term beneficial effects on oxygenat saturation in children with mild to moderate OSA but insufficient evidence on the effectiveness of AHI on OSA treatment. The duration of the study period may not have been sufficient given that the growth of the adenoids and tonsils continues during early childhood. Further studies are needed to better establish the effects of intranasal steroids on OSA and the benefits of use when both OSA and asthma occur simultaneously [59].

Avoiding allergens and environmental exposures to tobacco smoke and air pollutants can decrease the frequency and severity of asthma exacerbation [60]. At the same time, decreased environmental exposure to tobacco smoke and pollution will decrease upper airway inflammation and nasal obstruction decreasing the severity of upper airway obstruction for OSA patients [60,61].

Obesity is a risk factor which links asthma and OSA; weight loss is presumed to decrease severity of both conditions in some patients. Lifestyle interventions that are targeted to improve weight in obese adults with asthma have been shown to improve asthma control and asthma-related quality of life [62]. A review of lifestyle interventions suggested a more proactive comprehensive approach that simultaneously addresses diet, physical activity, and weight loss of at least 5–10% can improve overall asthma control [63]. Since weight loss decreases inflammation there is interest in evaluating if weight loss would also improve OSA symptoms and severity. Adolescents with OSA who underwent weight loss or bariatric surgery had a substantial improvement or resolution in the OSA; however, these studies are not generalizable to the population [64,65]. 

## 7. Conclusions

Asthma and OSA are common respiratory disorders in both the adult and pediatric populations. With the significant overlap in symptoms, risk factors, pathophysiology, and treatment in these 2 conditions, recent studies suggested an association with these two conditions. There is a reciprocal interaction, with asthma predisposing to OSA, and OSA worsening control and outcomes from asthma. Because of this association, understanding the mechanism of their relationship will help effectively treat both conditions. For patients with poorly controlled asthma, it is important to consider OSA as a factor contributing to exacerbations. Early recognition of the co-existence of OSA and asthma will improve the treatment outcome and may reduce morbidity and mortality as a result of these comorbidities. Further research is required to determine the optimal type and timing of treatment.

## Data Availability

Not applicable.

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
