# Peer review of "The Association between Asthma and OSA in Children"

_children, 2022, doi:10.3390/children9101430_

Round 1

Reviewer 1 Report

Comments to the authors: 

The manuscript presents a nice review on the bidirectional and interlaced association between OSA and asthma in pediatric population. It is a well-conducted summary of the evidence supporting overlapping pathophysiology, risk factors, and treatments. 

I underline here some minor corrections before considering for publication. 

Introduction

line 26: OSA needs to be spelt out as obstructive sleep apnea the first time it is used. 

lines 27-28: I would remove the sentence “and therefore it is important to study the interplay of these very common conditions in children”. This is normally a sentence that can be put at the end of a paragraph where you can justify the importance of the study. In this case, instead, it is the introductory passage of the whole manuscript, where the reader has no information yet. 

lines 31-32: does this prevalence refer to adults or children? As the topic is asthma in children, I would provide specific data only on children, to decrease the confusion of the reader and avoid going out of topic. Also, provide a reference to the sentence. 

lines 37-38: please, provide a reference. 

Common risk factors: 

In general, I suggest that this section of common risk factors should be moved after the pathophysiology. It is only a suggestion, as I find more explicative to present the pathophysiology first, with the explanation of the inflammation, hyperexcitation of upper airway, etc, to better understand the picture provided by the Common risk factors. 

In the first paragraph of the subsection “common risk factors”, the author lists some findings, such as early-onset wheezing, gender, to name a few. However, the paragraph is named as “common risk factors”, which suggests that the risk factors listed should be common for both. In the passage, it is clear that these factors are associated with the development of asthma. Are they also associated with OSA? in that case, references should be provided. Otherwise, I would omit the risk factors that are not supported from studies for OSA development.

Also, it does not make much sense to just mention few, but not all, of them in the introductory paragraph (such as obesity and GERD), and then repropose the same factors with a more detailed explanation later in the manuscript. I propose to mention ALL of them in the introductory paragraph (lines 61 to 73), or to omit them and to leave the discussion to the following subparagraph. 

Lines 99-100: provide the reference of the study. 

lines 104-105: please, elaborate the concept further. The fact that premature babies have impaired neuromuscular development means that asthma and OSA may be a consequence of an inappropriate neuromuscular control? This concept is very interesting, and might worth a deeper explanation based on pathophysiology of the two conditions. 

line 111: I would add, “predisposes the infant to longer term airway hyperreactivity, which is listed among the pathophysiology of both asthma and OSA” 

line 113: if environmental tobacco exposure is not repeated anymore throughout the manuscript, the acronym can be omitted. The term is repeated in line 305: just omit the acronym or use it consistently. 

line 116; line 124: please, provide a reference.  

line 122: spell out RSV 

The acronym of SDB (sleep disordered breathing) is introduced for the first time in line 172. However, the acronym has already been used earlier in the manuscript (e.g. line 132). Please, correct it. 

line 136: merge the concept expressed here with the sentence of lines 132-134, for a better readability from the reader. The two concepts written in this manner seem in contradiction. 

lines 145-149: here you are not talking about GERD anymore. This passage needs to be moved to the introductory paragraph of Common Risk Factors

line 190: provide different references to support the statement 

line 265: I believe “tonsillectomy and adenoidectomy” or “adenotonsillectomy” to be a more appropriate terms for the acronym of TA

line 270: what is “adenotonsillar”? I encourage the author to consistently use the acronym TA introduced earlier 

line 272: it is better to spell out ED in emergency department

lines 272-273: there is incorrect punctuation in the sentence

line 277: I assume B-agonist is beta-agonist. Please, change it or use the Greek alphabet. 

lines 189-291: what are the atopic factors the authors refer to? Please, explain. 

lines 302-303: I would be more caution in declaring that a causal relationship exists between asthma and OSA. One thing, as supported by the literature, is an association. If the authors here propose a “causal relationship”, what comes first? I would express it as a bidirectional relationship, or an association. However, none of the evidence points at causality so far. 

lines 305-308: are there any reference or studies to support this statement? 

Conclusion: remove the explanation of OSA as obstructive sleep apnea, as it should be introduced earlier in the manuscript and not repeated it anymore (line 311). 

Author Response

Dear Reviewer 1,

We value your time and consideration. The following edits have been made:

line 26: OSA needs to be spelt out as obstructive sleep apnea the first time it is used.

Thank you for your feedback, OSA spelled out

lines 27-28: I would remove the sentence “and therefore it is important to study the interplay of these very common conditions in children”. This is normally a sentence that can be put at the end of a paragraph where you can justify the importance of the study. In this case, instead, it is the introductory passage of the whole manuscript, where the reader has no information yet.

Thank you for this feedback. This sentence was removed.

lines 31-32: Does this prevalence refer to adults or children? As the topic is asthma in children, I would provide specific data only on children, to decrease the confusion of the reader and avoid going out of topic. Also, provide a reference to the sentence.

Thank you for your feedback, this section has been edited to pertain only to children.

lines 37-38: Please, provide a reference.

Thank you for this feedback, reference has been added.

Lines 99-100: Provide the reference of the study.

Reference 19 was added.

lines 104-105: Please, elaborate the concept further. The fact that premature babies have impaired neuromuscular development means that asthma and OSA may be a consequence of an inappropriate neuromuscular control? This concept is very interesting, and might worth a deeper explanation based on pathophysiology of the two conditions.

 Thank you for this feedback. We elaborated on this topic however no literature is available on the effect of prematurity on neuromuscular development predisposing to OSA or asthma.

line 111: I would add, “predisposes the infant to longer term airway hyperreactivity, which is listed among the pathophysiology of both asthma and OSA”

This sentence was added.

Line 113: if environmental tobacco exposure is not repeated anymore throughout the manuscript, the acronym can be omitted. The term is repeated in line 305: just omit the acronym or use it consistently.

Thank you for your feedback, the acronym was omitted.

line 122: spell out RSV

Thank you for your feedback, spelled out RSV.

line 124: provide a reference.

Thank you for this feedback, new reference has been added. (See new reference [106])

line 136: merge the concept expressed here with the sentence of lines 132-134, for a better readability from the reader. The two concepts written in this manner seem in contradiction. \

This sentence was modified.

Lines 145-149: here you are not talking about GERD anymore. This passage needs to be moved to the introductory paragraph of Common Risk Factors

This paragraph pertains to predisposition of male population for both asthma and OSA. The label Sex for this paragraph was left on the previous page. This paragraph is not part of GERD but under the label Sex.

lines 189-291: what are the atopic factors the authors refer to? Please, explain. 

Thank you for your feedback. We changed factors to diseases and added reference 103.

line 190: provide different references to support the statement

Reference 101 and 102 were added.

line 265: I believe “tonsillectomy and adenoidectomy” or “adenotonsillectomy” to be a more appropriate terms for the acronym of TA

Thank you for your feedback, changed the acronym to be more appropriate for TA.

line 270: what is “adenotonsillar”? I encourage the author to consistently use the acronym TA introduced earlier

Thank you for your feedback, we changed it to represent the acronym.

line 272: it is better to spell out ED in emergency department

Thank you for your feedback, changed the acronym.

lines 272-273: there is incorrect punctuation in the sentence

Thank you for your feedback, fixed the punctuation.

line 277: I assume B-agonist is beta-agonist. Please, change it or use the Greek alphabet.

Thank you for your feedback, fixed symbol.

lines 302-303: I would be more caution in declaring that a causal relationship exists between asthma and OSA. One thing, as supported by the literature, is an association. If the authors here propose a “causal relationship”, what comes first? I would express it as a bidirectional relationship, or an association. However, none of the evidence points at causality so far.

Thank you for this feedback, the word causal relationship was removed and changed to association.

lines 305-308: are there any reference or studies to support this statement?

Reference 104 and 105 were added.

Conclusion: remove the explanation of OSA as obstructive sleep apnea, as it should be introduced earlier in the manuscript and not repeated it anymore (line 311).

Thank you for this feedback, the explanation has been deleted.

Reviewer 2 Report

The authors have placed a significant amount of effort into compiling this data. Although numerous references are listed, the review could flow better. In addition, there are typing errors. For eg. line 270 page 6, line 107 page 3. 

Author Response

Dear Reviewer 2,

Thank you for your time and consideration. We have read your advice and edited for typing errors and improvement of flow.